# The Effect of the Histone Chaperones HSPA8 and DEK on Tumor Immunity in Hepatocellular Carcinoma

**DOI:** 10.3390/ijms24032653

**Published:** 2023-01-31

**Authors:** Chuanxin Yang, Yaodi Shao, Xiangjun Wang, Jie Wang, Puxiongzhi Wang, Chao Huang, Wei Wang, Jian Wang

**Affiliations:** 1Department of Hepatobiliary and Pancreatic Surgery, Shanghai Sixth People’s Hospital Affiliated to Shanghai Jiao Tong University School of Medicine, Shanghai 200233, China; 2Shanghai Diabetes Institute, Department of Endocrinology and Metabolism, Shanghai Sixth People’s Hospital Affiliated to Shanghai Jiao Tong University School of Medicine, Shanghai 200233, China; 3Department of Cell Biology, Medical School, Kunming University of Science and Technology, Kunming 650500, China

**Keywords:** hepatocellular carcinoma, histone chaperone, HSPA8, DEK, tumor immunity

## Abstract

Complex immune contexture leads to resistance to immunotherapy in hepatocellular carcinoma (HCC), and the need for new potential biomarkers of immunotherapy in HCC is urgent. Histone chaperones are vital determinants of gene expression and genome stability that regulate tumor development. This study aimed to investigate the effect of histone chaperones on tumor immunity in HCC. Bioinformatics analyses were initially performed using The Cancer Genome Atlas (TCGA) database, and were validated using the Gene Expression Omnibus (GEO) database and the International Cancer Genome Consortium (ICGC) database. Immune-related histone chaperones were screened with the Spearman rank coefficient. Consensus clustering was utilized to divide the HCC samples into two clusters. ESTIMATE, CIBERSORT and ssGSEA analyses were performed to assess immune infiltration. The expression of immunomodulatory genes, chemokines and chemokine receptors was analyzed to evaluate sensitivity to immunotherapy. The differentially expressed genes (DEGs) were included in weighted gene coexpression network analysis (WGCNA) to identify the hub genes. Enrichment analyses were used to investigate the functions of the hub genes. The Kaplan-Meier method and log-rank test were conducted to draw survival curves. A Cox regression analysis was utilized to identify independent risk factors affecting prognosis. *HSPA8* and *DEK* were screened out from 36 known histone chaperones based on their strongest correlation with the ESTIMATE score. Cluster 2, with high *HSPA8* expression and low *DEK* expression, tended to have stronger immune infiltration and better sensitivity to immunotherapy than Cluster 1, with low *HSPA8* expression and high *DEK* expression. Furthermore, WGCNA identified 12 hub genes closely correlated with immune infiltration from the DEGs of the two clusters, of which *FBLN2* was proven to be an independent protective factor of HCC patients. *HSPA8* and *DEK* are expected to be biomarkers for precisely predicting the effect of immunotherapy, and *FBLN2* is expected to be a therapeutic target of HCC.

## 1. Introduction

Worldwide, liver cancer is the fifth and seventh leading cause of cancer-related death in men and women, respectively, with an increasing incidence rate [1]. Because the early diagnosis of liver cancer is difficult, most patients are diagnosed in the advanced stage [2]. By that time, the treatment is difficult and has a poor curative effect, resulting in a five-year survival rate of only 20% [1].

Liver cancer includes hepatocellular carcinoma (HCC) and intrahepatic cholangiocarcinoma, and the former is the main pathological type of liver cancer [3]. With regard to treatment, chemotherapy, targeted therapy and immunotherapy are the therapeutic options for patients with advanced stage HCC because radical surgery is limited to HCC patients with early stage HCC [4]. In recent years, immune checkpoint inhibitors (ICIs), including pembrolizumab, nivolumab, durvalumab and atezolizumab have been used in HCC patients, but single-agent ICIs have shown disappointing results [5]. Conversely, immune-based combination therapy has become the first-line therapy for HCC [6,7]. For instance, the phase III IMbrave150 trial showed that the combination of ICI atezolizumab and antiangiogenic agent bevacizumab enabled unresectable HCC patients to acquire better clinical outcomes than single-agent sorafenib [8]. Although ICIs have shown promise for the treatment of HCC, several questions remain unanswered. Among these, the lack of validated biomarkers of response represents a major challenge since only some HCC patients benefit from immunotherapy [9]. Therefore, a greater understanding of the role of potential biomarkers is fundamental. In addition, the molecular mechanisms regulating immune responses and evasion remain unclear due to the complexity of the immune system [10]. Therefore, biomarkers to precisely predict the effect of immunotherapy in HCC need to be identified and validated.

Histone chaperones are vital determinants of histone fate which regulate the disassembly, reassembly and repositioning of nucleosomes to spatially and temporally control histone accessibility for transcription factors and RNA polymerases [11]. Histone chaperones play a major role in gene expression and genome stability; however, dysfunctions of histone chaperones are associated with a higher risk of various diseases including cancers [12]. For example, the facilitates chromatin transcription complex, which consists of the histone chaperones SUPT16H and SSRP1, has been proven to promote oxidative stress adaptation and to be a potential target for the treatment of HCC [13].

The histone chaperone *HSPA8/HSC70* is a member of the heat shock protein family that is vital to biological processes, including autophagy and immunity [14]. Additionally, *HSPA8* has been reported to be highly immunogenic and overexpressed in several tumor cells, which indicates its close relationship with tumors [15]. Specifically, previous studies have demonstrated that *HSPA8* is associated with tumor metastasis and recurrence in HCC [16,17]. Another histone chaperone, *DEK*, a tumor-promoting factor, has been reported to potentially create an immune-suppressed tumor microenvironment [18]. Specifically, this finding was confirmed because *DEK* loss induced an inflammatory or immune response by activating the NF-κB pathway [19]. Furthermore, a recent study showed that *DEK* influences tumor cell migration and invasion in HCC [20]. However, the relationship between the two histone chaperones and tumor immunity in HCC remains unclear.

To investigate the effect of histone chaperones on tumor immunity in depth, we performed an integrated bioinformatics analysis of gene expression data of HCC. Surprisingly, we screened *HSPA8* and *DEK* from 36 known histone chaperones [11,13] and divided HCC samples into two clusters based on the expression matrix of *HSPA8* and *DEK*. We then found that Cluster 2, with high *HSPA8* expression and low *DEK* expression, tended to have stronger immune infiltration and better sensitivity to immunotherapy than Cluster 1, with low *HSPA8* expression and high *DEK* expression. We also identified 12 hub genes from the differentially expressed genes (DEGs) of the two clusters related to immune infiltration. In particular, *FBLN2*, one of the hub genes, is an independent protective factor in HCC patients and is associated with tumor immunity. In conclusion, *HSPA8* and *DEK* greatly affect the tumor immunity of HCC and could potentially be regarded as biomarkers for precisely predicting the effect of immunotherapy in HCC. Furthermore, *FBLN2* could be a therapeutic target of HCC. Importantly, fibulin-2, which is encoded by *FBLN2*, is a secreted protein that has been commercially prepared. Therefore, immunotherapy combined with recombinant fibulin-2 or activators of *FBLN2* may bring new hope for patients with advanced HCC.

## 2. Results

### 2.1. Screening of Immune-Related Histone Chaperones and Consensus Clustering of the Cancer Genome Atlas-Liver Hepatocellular Carcinoma (TCGA-HCC) Samples

To investigate the potential impact of histone chaperones on tumor immunity in HCC, we investigated the correlation between 36 known histone chaperones [11,13] and the ESTIMATE score in 369 TCGA-HCC samples (Figure 1A). In view of the highest absolute value of the correlation with the ESTIMATE score, *HSPA8* and *DEK* were screened out for subsequent analyses. Next, we analyzed the expression of *HSPA8* and *DEK* between 369 TCGA-HCC samples and 50 normal samples. The results showed that the expression of both genes was higher in tumor tissues than in normal tissues (Figure 1B,C). We also investigated the protein level of HSPA8 and DEK in human HCC samples via the Human Protein Atlas (HPA) database, and found that the protein level of both HSPA8 and DEK was higher in HCC tissues than in normal liver tissues (Figure 1D,E). We then explored the correlation between the expression of the above two genes and the abundance of tumor-infiltrating lymphocytes (TILs) in TISIDB and found that the expression of *HSPA8* was positively correlated with the abundance of TILs in HCC, whereas the expression of *DEK* showed the opposite trend (Figure 1F,G). Afterward, consensus clustering was conducted on TCGA-HCC samples based on the expression matrix of *HSPA8* and *DEK*, and the samples were ultimately divided into two clusters (Figure 1H). As shown in the heatmap, Cluster 1 (n = 206) had low *HSPA8* expression and high *DEK* expression, whereas Cluster 2 (n = 163) had high *HSPA8* expression and low *DEK* expression. Considering the opposite expression profiles of *HSPA8* and *DEK* in the two clusters, a Spearman rank correlation analysis between the expression of *HSPA8* and *DEK* was performed in TCGA-HCC samples. However, an extremely weak, positive correlation (R = 0.19, *p* = 0.00024) was found (Figure 1I).

### 2.2. Comparison of Immune Infiltration between the Two Clusters

To obtain a comprehensive understanding of tumor immunity in the two clusters, the four ESTIMATE indices were calculated and Cluster 2 was found to have a higher stromal score, immune score and ESTIMATE score, but lower tumor purity than Cluster 1 (Figure 2A–D). In addition, a CIBERSORT analysis showed differences in the proportion of partial immune cells between the two clusters (Figure 2E). Importantly, a single-sample gene set enrichment analysis (ssGSEA) demonstrated that Cluster 2 had a higher expression of 24 immune cell subtypes (e.g., activated and immature B cells, activated and memory CD8^+^ T cells, activated and immature dendrite cells, natural killer cells, and natural killer T cells) than Cluster 1 (Figure 2F). Moreover, the expression of major histocompatibility complexes (MHCs) in Cluster 2 tended to be higher than that in Cluster 1 (Figure 2G). Taken together, Cluster 2 tended to have stronger immune infiltration than Cluster 1.

### 2.3. Evaluation of Sensitivity to Immunotherapy of the Two Clusters

To explore the potential sensitivity to immunotherapy, we compared the expression of several immunomodulatory genes between the two clusters, including *PD1*-related genes, *CTLA4*-related genes, and other antagonists or agonists of T-cell activation [21,22,23]. The results showed that most of the changed genes were upregulated in Cluster 2 (Figure 3A–D). Furthermore, chemokines play an important role in the activation of immune cells, and the same chemokine may play opposite roles in different tumors [24]. We analyzed the expression of known chemokines and chemokine receptors that inhibit or promote the progression of HCC [25] and found that Cluster 2 had higher expression of anti-HCC chemokines and chemokine receptors (*CCL4*, *CCL19*, *CCL21*, *CCR5*, *CCR7*) and lower expression of pro-HCC chemokines and chemokine receptors (*CCL28* and *CCR10*) than Cluster 1 (Figure 3E,F). Because gene mutations also affect sensitivity to immunotherapy [26], we evaluated the gene mutation profile and drew the landscapes of the two clusters (Appendix A). However, tumor mutation burden (TMB) did not significantly differ between the two clusters (Appendix A). The above results indicated that Cluster 2 may have better sensitivity to immunotherapy than Cluster 1.

### 2.4. Screening of Hub Genes Related to Immune Infiltration between the Two Clusters

To explain the difference in immune features between the two clusters, we compared the gene expression profiles of the two clusters and identified 505 DEGs (197 upregulated and 308 downregulated, Cluster 2 vs. Cluster 1) (Figure 4A). The DEGs were then subjected to a weighted correlation network analysis (WGCNA) to screen the genes that had the strongest correlation with tumor immunity. We set the soft threshold at 3 to ensure that the constructed coexpression network approached scale-free distribution (Figure 4B, Appendix A). The DEGs were then divided into different modules with colors. Ultimately, we found that the blue module with 121 genes was strongly associated with tumor immunity because it had the highest stromal, immune and ESTIMATE scores and the lowest tumor purity (Figure 4C).

Next, to confirm the mechanism of 121 genes related to tumor immunity, we performed Gene Ontology (GO) and the Kyoto Encyclopedia of Genes and Genomes (KEGG) enrichment analyses. A GO enrichment analysis showed that the 121 genes were enriched in immune-related terms including dendritic cell dendrite assembly, regulation of dendritic cell dendrite assembly, CCR7 chemokine receptor binding, cytokine activity and chemokine activity through GO enrichment analysis (Figure 4D). In addition, the results of KEGG analysis showed that these genes were involved in immune-related pathways, including cytokine-cytokine receptor interaction, TGF-β signaling pathway, viral protein interaction with cytokine and cytokine receptor and chemokine signaling pathway (Figure 4E). Moreover, we selected 12 hub genes (PODN, DCN, LUM, CCDC80, SVEP1, AEBP1, FBLN2, TNXB, HAND2, INMT, OMD, and GPBAR1) that were upregulated in Cluster 2 to specify the key genes of the 121 genes (Figure 4F, Appendix A).

### 2.5. Determination of FBLN2 Functions

First, we constructed a protein-protein interaction (PPI) network of the 12 hub genes, and found interactions among FBLN2, OMD, LUM and DCN (Figure 5A). A Spearman’s rank correlation analysis showed moderate to strong correlations among the 12 hub genes (Figure 5B). Next, ESTIMATE and ssGSEA analyses indicated that these genes closely correlated with immune infiltration (Figure 5C,D). Furthermore, we evaluated the correlation between the 12 hub genes and the prognosis of TCGA-HCC patients individually and found that only *FBLN2* was associated with prognosis. TCGA-HCC patients with high *FBLN2* expression had significantly longer overall survival (OS) than those with low *FBLN2* expression (Figure 6A). In addition, a multivariate Cox regression analysis showed that *FBLN2* was an independent protective factor in TCGA-HCC patients (Figure 6B). We further explored the effect of *FBLN2* on tumor immunity through TISIDB and found that the expression of *FBLN2* was positively correlated with the abundance of TILs in HCC (Figure 6C), and patients with high *FBLN2* expression had stronger immune infiltration than those with low *FBLN2* expression based on a CIBERSORT analysis and ssGSEA (Figure 6D,E).

### 2.6. Validation of Immune Features by the Gene Expression Omnibus (GEO) and International Cancer Genome Consortium (ICGC) Databases

The GEO and ICGC databases were used as verification databases to validate previous results from the TCGA database. As previously described, we divided 164 HCC samples from GSE19977 [27,28] into two clusters based on the expression matrix of *HSPA8* and *DEK* via consensus clustering (Appendix A). Interestingly, a Spearman’s rank correlation analysis showed a weak, negative correlation (R = −0.31, *p* = 6 × 10^−5^) between the expression of *HSPA8* and *DEK* (Appendix A). The ESTIMATE, CIBERSORT and ssGSEA analyses showed similar results to those in the TCGA database. Cluster 2, with high *HSPA8* expression and low *DEK* expression, was found to have a higher stromal score, immune score, ESTIMATE score and expression of 17 immune cell subtypes than Cluster 1, with low *HSPA8* expression and high *DEK* expression. (Appendix A). In addition, the expression of MHCs, immunomodulatory genes, chemokines and chemokine receptors of the two clusters was evaluated as before. Cluster 2 had a higher expression of MHCs, immunomodulatory genes and anti-HCC chemokines and chemokine receptors (Appendix A).

Similarly, we divided 240 HCC samples from ICGC-Liver Cancer-RIKEN, JP (LIRI-JP) into two clusters in the same manner, and the remaining analyses (Spearman rank correlation, ESTIMATE, CIBERSORT and ssGSEA analyses as well as the expression of MHCs, immunomodulatory genes, chemokines and chemokine receptors) yielded approximate results (Appendix A). Cluster 2, with high *HSPA8* expression and low *DEK* expression, was found to have a higher stromal score, immune score, ESTIMATE score, expression of 18 immune cell subtypes, expression of MHCs, immunomodulatory genes and anti-HCC chemokines and chemokine receptors than Cluster 1, with low *HSPA8* expression and high *DEK* expression. Taken together, Cluster 2 is characterized by stronger tumor immunity than Cluster 1 in the TCGA, GEO and ICGC databases.

### 2.7. Verification of the Expression of HSPA8 and DEK in HCC Cells

We compared the expression of *HSPA8* and *DEK* in human HCC cell lines (HepG2 and Huh-7) with that in human normal hepatocytes (L02) through quantitative real-time polymerase chain reaction (qRT-PCR) and western blotting. The results showed that HCC cells had higher mRNA and protein expression of *HSPA8* and *DEK* than L02 cells (Figure 7).

## 3. Discussion

The complexity of the immune system leads to resistance to immunotherapy in HCC, and the lack of biomarkers for precisely predicting the effect of immunotherapy in HCC remains a major unsolved challenge. In addition, clinical trials on HCC immunotherapy have widely differed in terms of drugs, patients, designs, study phases, and inconsistent clinical outcomes. Therefore, applicable and unified biomarkers could help reduce the bias of clinical trials. Histone chaperones are vital determinants of gene expression and genome stability that regulate tumor development. However, the relationship between histone chaperones and tumor immunity in HCC remains unclear. In the current study, we revealed that the histone chaperones *HSPA8* and *DEK* strongly influence the tumor immunity of HCC. HCC patients with high *HSPA8* expression and low *DEK* expression tend to have stronger immune infiltration and better sensitivity to immunotherapy. Moreover, *FBLN2*, one of the hub DEGs, is an independent protective factor and is associated with immune infiltration in HCC. Therefore, *HSPA8* and *DEK* are expected to be biomarkers for precisely predicting the effect of immunotherapy in HCC, and *FBLN2* is expected to be a therapeutic target of HCC.

First, we found that *HSPA8* and *DEK* had the strongest correlation with the ESTIMATE score among 36 known histone chaperones, indicating that these histone chaperones are most closely related to tumor immunity in HCC. Significantly, *HSPA8* was positively correlated with the ESTIMATE score, whereas *DEK* was negatively correlated with the ESTIMATE score, which is in line with previous studies [14,15,18,19].

Next, we utilized consensus clustering to divide HCC patients into two clusters, and we compared immune infiltration between the two clusters using ESTIMATE, CIBERSORT and ssGSEA. In the ESTIMATE analysis, Cluster 2 had a higher stromal score, immune score and ESTIMATE score but lower tumor purity than Cluster 1. The CIBERSORT analysis identified differences in the proportion of partial immune cells between the two clusters. In the ssGSEA, Cluster 2 had higher expression of 24 immune cell subtypes than Cluster 1, including CD8^+^ T cells, dendrite cells (DCs), natural killer (NK) cells and natural killer T (NKT) cells. Specifically, the current checkpoint inhibitor immunotherapy works mainly by blocking the binding between PD-L1 on the surface of tumor cells and PD-1 on the surface of CD8^+^ T cells [29], and CD8^+^ T cells were proven to be a favorable prognostic biomarker for HCC [30]. DCs are key antigen-presenting cells with a role in initiating and regulating antitumor immunity [31]. NK cells have the capacity to promote antitumor immunity by enhancing antibody and T-cell responses [32]. Previous studies reported that NKT cells mediate liver-selected tumor inhibition induced by the gut microbiome [33], and enhance antitumor immunity by reinvigorating exhausted CD8^+^ T cells in checkpoint inhibitor immunotherapy [34]. Taken together, the above three analyses confirmed that Cluster 2 possessed stronger immune infiltration than Cluster 1.

Furthermore, immunomodulatory genes were proven to be predictive biomarkers of immunotherapy. Patients with a higher expression of *PD-L1* and *PD-L2* reportedly will benefit more from immunotherapy [35,36]. Here, we compared the expression of several immunomodulatory genes between the two clusters, and the results showed that most of the changed genes were upregulated in Cluster 2. Moreover, we found that Cluster 2 had a higher expression of anti-HCC chemokines and chemokine receptors, and a lower expression of pro-HCC chemokines and chemokine receptors than Cluster 1. Thus, our results strongly suggest that Cluster 2 has better sensitivity to immunotherapy than Cluster 1.

Subsequently, we utilized WGCNA to determine the potential reason for the difference in immune features between the two clusters. The DEGs in the blue module had the strongest correlation with tumor immunity, which was further confirmed by GO and KEGG enrichment analysis. We then screened 12 hub genes from the blue module based on MM and GS, such as *PODN*, *DCN*, *CCDC80*, *SVEP1* and *AEBP1*, which were reported to be predictive biomarkers and correlated with immune infiltration in various cancers [37,38,39,40,41]. In addition, ESTIMATE and ssGSEA analyses indicated that these 12 hub genes closely correlated with immune infiltration, and all were upregulated in Cluster 2. Therefore, these 12 hub genes might explain why Cluster 2 possesses stronger immune infiltration and better sensitivity to immunotherapy than Cluster 1.

Intriguingly, *FBLN2* was the only hub gene associated with the prognosis of TCGA-HCC patients. *FBLN2* encodes an extracellular matrix protein, fibulin-2, which was reported to suppress the proliferation of non-small cell lung cancer and the metastasis of breast cancer [42,43]. However, studies involving the relationship between fibulin-2 and tumor immunity are lacking. In the current study, Cluster 2 had higher *FBLN2* expression than Cluster 1, and patients with high *FBLN2* expression presented longer OS than those with low *FBLN2* expression. In addition, the expression of *FBLN2* was positively correlated with the abundance of TILs and immune infiltration in HCC patients. Therefore, *FBLN2* has the potential to be a therapeutic target of HCC. Importantly, fibulin-2 is commercially available. Therefore, immunotherapy combined with recombinant fibulin-2 or activators of *FBLN2* is a promising research direction.

Although the current study reveals the effect of the histone chaperones *HSPA8* and *DEK* on tumor immunity in HCC, it has certain limitations and drawbacks. First, our results were primarily obtained by bioinformatics analyses, which require laboratory-based experiments for further confirmation. Second, because our study cohorts were collected from different public databases, intratumor or intrapatient tumor heterogeneity was inevitable. Therefore, further studies to unravel the specific mechanisms regulating tumor immunity should be performed.

## 4. Materials and Methods

### 4.1. Data Acquisition

The transcriptome profiling data of 369 HCC samples and the corresponding clinical information as well as their mutation annotation file (MAF) files containing somatic mutation information were downloaded from the TCGA database. The transcriptome profiling data of 164 HCC samples were downloaded from the GEO database (GSE19977). The transcriptome profiling data of 240 HCC samples were downloaded from the ICGC database (ICGC-LIRI-JP).

### 4.2. ESTIMATE Analysis

After the transcriptome profiling data were imported into R software (version 4.1.3), the R package “estimate” [44] was used, and the ESTIMATE analysis was used to evaluate the tumor microenvironment of each patient with HCC. The results of ESTIMATE analysis included the stromal score, immune score, ESTIMATE score and tumor purity of each HCC patient. The stromal score and immune score represented the abundance of stromal cells and immune cells, respectively. The ESTIMATE score was calculated based on the stromal score and immune score, representing the extent of immune infiltration. The tumor purity equaled one minus the ESTIMATE score, representing the abundance of tumor cells in tumor tissue.

### 4.3. CIBERSORT Analysis

After the transcriptome profiling data were imported into R software, the code “CIBERSORT” [45] was used, and then a CIBERSORT analysis was conducted to estimate the abundance of 22 immune cells in each HCC sample. The ordinate of the result represents the percentage of 22 immune cells.

### 4.4. ssGSEA

After the transcriptome profiling data were imported into R software, the R package “GSVA” [46] was used, and then ssGSEA was applied to quantify the extent of immune infiltration in each HCC sample. The ordinate of its result represented the extent of infiltration of 28 immune cells.

### 4.5. HPA Database

The Human Protein Atlas (HPA) database (https://www.proteinatlas.org/, accessed on 14 May 2022) was used to investigate the protein content in human HCC samples and normal liver samples. The HPA database is a web portal that prestores the tissue distribution information of 26,000 human proteins in tumor tissues, including HCC. Once we inputted a protein name (e.g., *HSPA8*, *DEK*), we automatically obtained the immunohistochemical results of this protein in HCC samples and normal liver samples.

### 4.6. TISIDB

TISIDB (http://cis.hku.hk/TISIDB/index.php, accessed on 9 May 2022) [47] was used to assess the correlation between gene expression and the abundance of TILs in HCC samples. TISIDB is a web portal that prestores multiple heterogeneous data types. Once we inputted a gene name (e.g., *HSPA8*, *DEK* or *FBLN2*), we automatically generated a heatmap of the abundance of TILs based on the expression of this gene.

### 4.7. Consensus Clustering

After the transcriptome profiling data were imported into the R software, the gene names (e.g., “HSPA8” and “DEK”) were set as the clustering criteria. The R package “ConsensusClusterPlus” [48] was then used, and consensus clustering was used to divide the HCC samples into different clusters based on the expression matrix of *HSPA8* and *DEK*. The result was a heatmap of the expression of *HSPA8* and *DEK* in each HCC sample, and the samples with the same expression matrix of *HSPA8* and *DEK* were assigned to the same cluster.

### 4.8. Tumor Mutation Burden Calculation

After the transcriptome profiling data and the corresponding MAF files containing somatic mutation information were imported into the R software, the R package “maftools” [49] was used, and then the TMB was calculated according to the preset algorithms. The value of TMB reflected the ability of the tumor to produce new antigens and predicted the sensitivity of immunotherapy.

### 4.9. DEG Identification

After the transcriptome profiling data were imported into the R software, |log_2_FoldChange| > 1 and adjusted *p* value < 0.05 were set as threshold values. Then, the R package “DESeq2” [50] was used, and the differential expression analysis was used to identify DEGs among the different clusters, which were generated by consensus clustering (see Section 4.6).

### 4.10. WGCNA

After the transcriptome profiling data and the corresponding clinical information were imported into the R software, the R package “WGCNA” [51] was used. The WGCNA was then conducted to divide DEGs (see Section 4.8) into several modules based on their correlations with the immune infiltration indices including stromal score, immune score, ESTIMATE score and tumor purity. The soft threshold was set according to the scale-free topology standard. The modules were named with different colors. The module with the strongest correlation with immune infiltration was then screened out. Subsequently, the hub genes in the module with the strongest correlation with immune infiltration were screened based on the threshold values of MM > 0.8 and GS > 0.6.

### 4.11. GO Enrichment Analysis

After all the gene names in the module with the strongest correlation with immune infiltration (see Section 4.9) were imported into the R software, the R package “clusterProfiler” [52] was used, and then a GO enrichment analysis was applied to investigate the biological functions of these genes. The results consisted of biological process (BP), cellular component (CC), and molecular function (MF), representing the potential biological functions of these genes.

### 4.12. KEGG Pathway Analysis

After all the gene names in the blue module with the strongest correlation with immune infiltration (see Section 4.9) were imported into R software, the R package “clusterProfiler” [52] was used, and a KEGG pathway analysis was applied to investigate the biological functions of these genes. The results showed the bubble chart of potential signaling pathways.

### 4.13. STRING

STRING (https://string-db.org/, accessed on 9 May 2022) [53] was used to construct the PPI network of the hub genes screened by WGCNA (see Section 4.9). STRING is a web portal for searching known protein interactions and predicting unknown protein interactions. Once we inputted the list of 12 hub gene names, we automatically generated the PPI network of these genes.

### 4.14. Cell Culture and qRT-PCR

The human HCC cell lines HepG2 and Huh-7, and the human normal hepatocyte line L02 were purchased from the Cell Bank of the Type Culture Collection of the Chinese Academy of Sciences. HepG2 and Huh-7 cells were cultured in Dulbecco’s Modified Eagle’s Medium (Gibco, Grand Island, NY, USA). L02 cells were cultured in Roswell Park Memorial Institute 1640 medium (Gibco). All media were supplemented with 10% fetal bovine serum (Gibco) and penicillin-streptomycin. All cultures were maintained in a humidified chamber with 5% CO_2_ at 37 °C. Total RNA was extracted using TRIzol reagent (Invitrogen, Carlsbad, CA, USA). cDNA was reverse transcribed by a HiScript 1st Strand cDNA Synthesis Kit (Vazyme, Nanjing, China). qPCR was performed using AceQ qPCR SYBR Green Master Mix (Vazyme). *18S* was selected as the internal control. The primer sequences were as follows:*18S-F,* 5′-TTCGAACGTCTGCCCTATCAA-3′;*18S-R,* 5′-ATGGTAGGCACGGCGACTA-3′;*HSPA8*-F, 5′-GCTTCTATCCAGAGGAGGTGTCTT-3′;*HSPA8*-R, 5′-GACCAGCAATAGTTCCAGCATCTT-3′;*DEK*-F, 5′-CTGGAATGGCAAGGAAGGCTAAG-3′;*DEK*-R, 5′-TTTGGTGGCTCCTCTTCACTTTC-3′.

### 4.15. Western Blotting

Cells were lysed in radioimmunoprecipitation lysis buffer (50 mM Tris, pH 7.4, 150 mM NaCl, 1% NP-40, 0.1% sodium dodecyl sulfate (SDS), 0.1% sodium deoxycholate) containing proteinase inhibitor cocktail (MedChemExpress, Monmouth Junction, NJ, USA). The protein samples were electrophoresed through 10% SDS polyacrylamide gels and transferred onto polyvinyl difluoride membranes (Millipore, Billerica, MA, USA). After blocking with 5% bovine serum albumin at room temperature for 1 h, the membranes were incubated with primary antibodies at 4 °C overnight, and then incubated with secondary antibodies at room temperature for 1 h. Immunoreactivity was detected with enhanced chemiluminescent autoradiography (Millipore). Chemiluminescence was determined using the AI600 System (GE Healthcare, Little Chalfont, Buckinghamshire, UK). The antibody against GAPDH (60004-1-Ig, dilution 1:1000) was purchased from Proteintech. Antibodies against HSPA8 (8444, dilution 1:1000) and DEK (29812, dilution 1:1000) were purchased from Cell Signaling Technology (Danvers, MA, USA).

### 4.16. Statistical Analysis

Normally distributed continuous variables were compared using the Student’s *t*-test; otherwise, they were compared using the Mann-Whitney U test. Categorical variables were analyzed using the chi square test or Fisher’s exact test, as appropriate. Correlations were analyzed with the Spearman rank coefficient. A binary logistic regression analysis was utilized to analyze the difference in histological grade between the two clusters. The Kaplan-Meier method and log-rank test were conducted to draw survival curves. A Cox regression analysis was utilized to identify independent risk factors affecting the prognosis. Boxplots are used to present the data between different groups. The middle line of the box shows the median value of the data; the upper and lower limits of the box are the 75% and 25% quartiles of the data, and the two vertical lines above and below the box represent the maximum and minimum values. A *p* value < 0.05 was considered statistically significant. All statistical tests were performed using R (version 4.1.3, The University of Auckland, Auckland, New Zealand) and SPSS (version 22.0, IBM, Armonk, NY, USA) software.

## 5. Conclusions

In conclusion, the histone chaperones *HSPA8* and *DEK* are closely related to the tumor immunity of HCC. By clustering HCC patients from TCGA-HCC, GSE19977 and ICGC-LIRI JP based on the expression matrix of *HSPA8* and *DEK*, we demonstrated that HCC patients with high *HSPA8* expression and low *DEK* expression tend to have stronger immune infiltration and better sensitivity to immunotherapy. The above phenomenon may be explained by 12 hub genes that were upregulated in Cluster 2 and that are strongly related to tumor immunity. Furthermore, *FBLN2*, one of the hub genes, is an independent protective factor in HCC patients and is associated with tumor immunity. Therefore, *HSPA8* and *DEK* are expected to be biomarkers for precisely predicting the effect of immunotherapy, and *FBLN2* could be a therapeutic target of HCC. Our study provides a new direction for discovering potential biomarkers of immunotherapy in HCC.

## Figures and Tables

**Figure 1 ijms-24-02653-f001:**
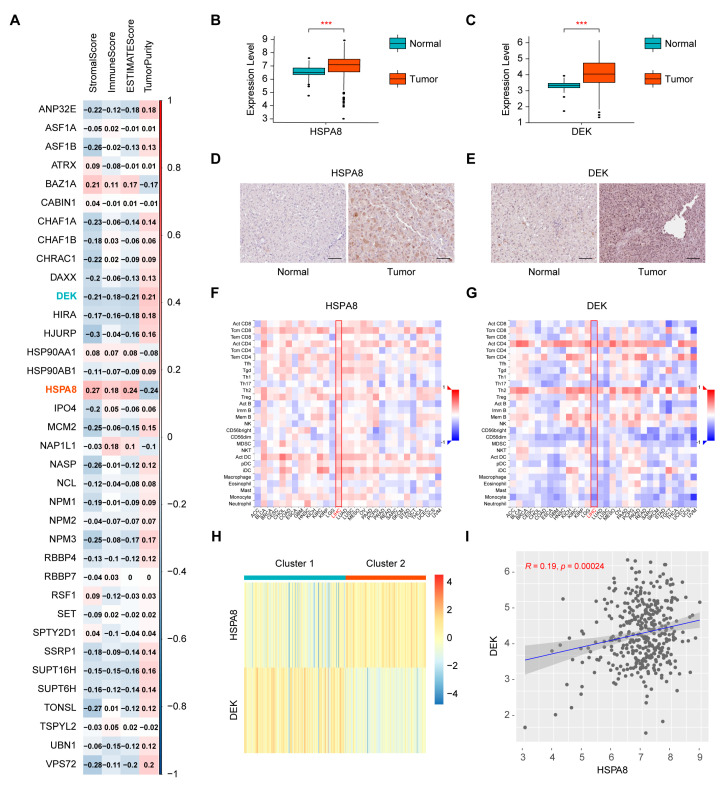
Screening of immune-related histone chaperones and clustering of 369 TCGA-HCC samples based on *HSPA8* and *DEK*. (**A**) Correlation analysis between 36 known histone chaperones and the four ESTIMATE indices, including stromal score, immune score, ESTIMATE score and tumor purity. (**B**,**C**) Comparison of the mRNA expression of *HSPA8* or *DEK* between tumor and normal tissues from the TCGA database. (**D**,**E**) Comparison of the protein level of HSPA8 or DEK between tumor and normal tissues from the HPA database. Scale bar, 100 µm. (**F**,**G**) Correlation analysis between *HSPA8* or *DEK* and the abundance of tumor-infiltrating lymphocytes by TISIDB. (**H**) TCGA-HCC samples are divided into two clusters according to *HSPA8* and *DEK* via consensus clustering. (**I**) Spearman rank correlation analysis between the expression of *HSPA8* and *DEK*. *** *p* < 0.001.

**Figure 2 ijms-24-02653-f002:**
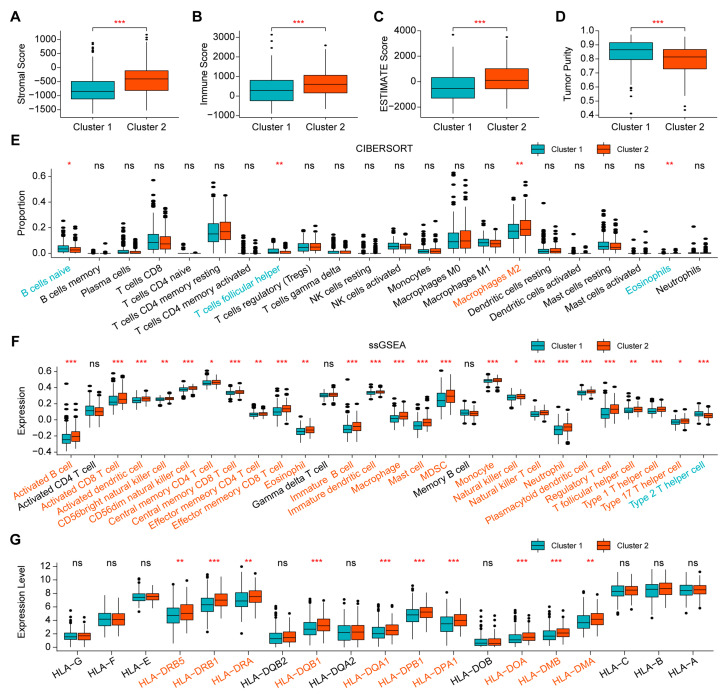
Comparison of immune infiltration between the two clusters. (**A**–**D**) Comparison of the four ESTIMATE indices, including stromal score, immune score, ESTIMATE score and tumor purity. (**E**) Comparison of the proportion of immune cells by CIBERSORT. (**F**) Comparison of the expression of immune cells by ssGSEA. (**G**) Comparison of the expression of MHCs. NS: no significance, * *p* < 0.05, ** *p* < 0.01, *** *p* < 0.001.

**Figure 3 ijms-24-02653-f003:**
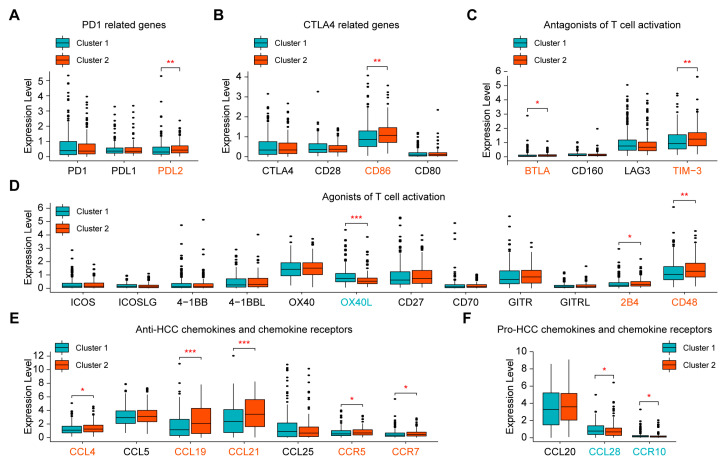
Evaluation of sensitivity to immunotherapy of the two clusters. (**A**–**D**) Comparison of the expression levels of immunomodulatory genes between Cluster 1 and Cluster 2. (**E**,**F**) Comparison of the expression levels of known chemokines and chemokine receptors that regulate the progression of HCC between Cluster 1 and Cluster 2. * *p* < 0.05, ** *p* < 0.01, *** *p* < 0.001.

**Figure 4 ijms-24-02653-f004:**
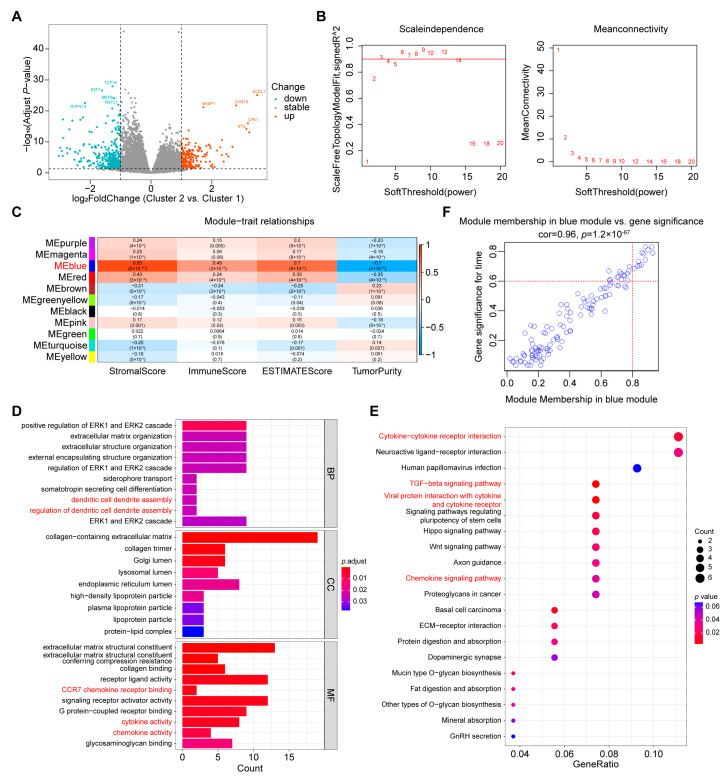
Screening of hub genes related to immune infiltration. (**A**) Volcano plot of DEGs. The red and blue dots represent upregulated and downregulated genes in Cluster 2, respectively. (**B**) Analysis of network topology for soft powers, and the soft threshold was set at 3. (**C**) Heatmap analysis between different modules and the four indices of ESTIMATE via WGCNA. (**D**,**E**) GO and KEGG enrichment analysis of the blue module, and immune-related GO terms or KEGG pathways are marked in red. (**F**) Scatter plot analysis of 121 genes in the blue module via WGCNA.

**Figure 5 ijms-24-02653-f005:**
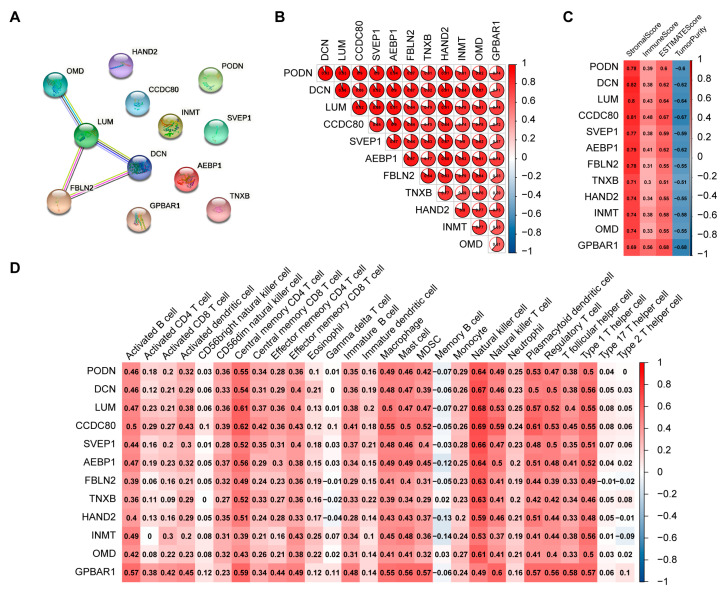
Determination of the functions of hub genes. (**A**) PPI network of hub genes. (**B**) Spearman rank correlation analysis among hub genes. (**C**) Correlation between hub genes and the stromal score, immune score, ESTIMATE score, and tumor purity of ESTIMATE. (**D**) Correlation analysis between hub genes and the expression of immune cells by ssGSEA.

**Figure 6 ijms-24-02653-f006:**
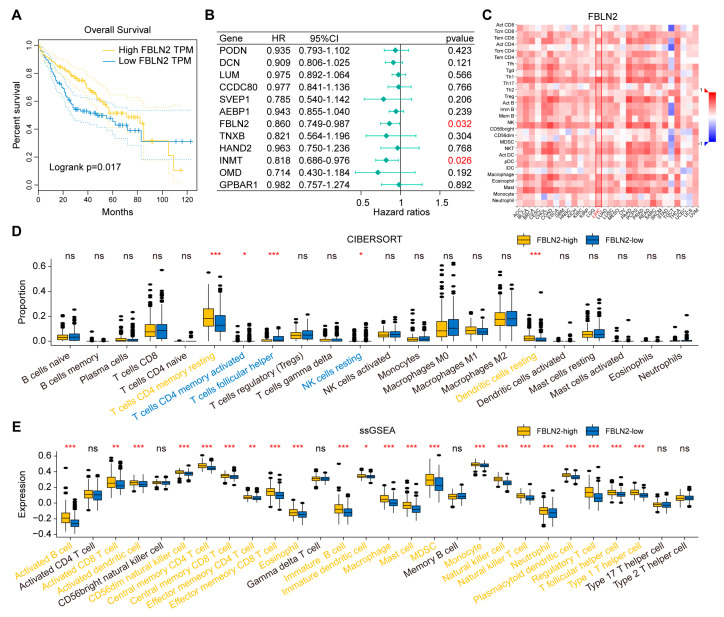
Determination of the functions of *FBLN2*. (**A**) Survival curves of patients with different expression levels of *FBLN2*. (**B**) Multivariate Cox regression analysis of the 12 hub genes. (**C**) Correlation analysis between *FBLN2* and the abundance of tumor-infiltrating lymphocytes through TISIDB. (**D**) Comparison of the proportion of immune cells by CIBERSORT in patients with different expression levels of *FBLN2*. (**E**) Comparison of the expression of immune cells by ssGSEA in patients with different expression levels of *FBLN2*. ns, no significance, * *p* < 0.05, ** *p* < 0.01, *** *p* < 0.001.

**Figure 7 ijms-24-02653-f007:**
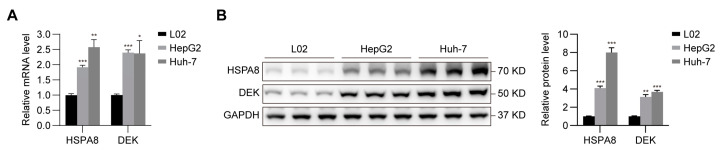
Verification of the expression of *HSPA8* and *DEK* in HCC cells using qRT-PCR (**A**) and western blotting (**B**). The results are representative of three separate experiments. * *p* < 0.05, ** *p* < 0.01, *** *p* < 0.001.

## Data Availability

The data of our study was derived from the public databases, and the original code can be obtained from the corresponding author upon reasonable request.

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
