# Peer review of "The Effect of the Histone Chaperones HSPA8 and DEK on Tumor Immunity in Hepatocellular Carcinoma"

_ijms, 2023, doi:10.3390/ijms24032653_

Round 1

Reviewer 1 Report

The authors have done an extensive bioinformactics analysis to investigate the effect of histone chaperones on tumor immunity in HCC using public databases. They revealed two histone chaperones HSPA8 and DEK from 36 known histone chaperones to be prognotic markers of HCC. The conclusions are well supported by the results. I have affermative position for the manuscript, however I feel that the number of figures should be reduced to 7 main figures with key message and remaining should be placed in supplementary. Other recommendation is that the methods section sounds superficial and it should be enriched with details.

Author Response

Reviewer 1:

The authors have done an extensive bioinformatics analysis to investigate the effect of histone chaperones on tumor immunity in HCC using public databases. They revealed two histone chaperones HSPA8 and DEK from 36 known histone chaperones to be prognostic markers of HCC. The conclusions are well supported by the results. I have affirmative position for the manuscript; however I feel that the number of figures should be reduced to 7 main figures with key message and remaining should be placed in supplementary. Other recommendation is that the methods section sounds superficial and it should be enriched with details.

A: Thank you very much for your positive feedback of our study. According to your suggestions, we have placed the previous Figures 7-10 in the Supplementary Files so that the number of figures has been reduced to 7 main figures. Moreover, we have comprehensively revised the Methods section to make it more detailed and understandable (line 406 – line 559).

Reviewer 2 Report

The study assesses a current, timely topic in HCC.
We recommend some changes:
- We believe this article is suitable for publication in the journal although major revisions are needed. The main strengths of this paper are that it addresses an interesting and very timely question and provides a clear answer, with some limitations. The authors should further speculate regarding how this study may impact on clinical practice and its translational implications.

- A linguistic revision should be performed by a professional service since there are some oversights and grammar mistakes. 
Immune checkpoint inhibitors (ICIs) including pembrolizumab, nivolumab, durvalumab, atezolizumab, etc. have been recently evaluated in HCC patients, and clinical trials assessing single-agent ICI have reported disappointing results. Conversely, immune-based combinations have been more striking. In fact, the phase III IMbrave150 trial assessing the combination of the antiangiogenic agent bevacizumab plus the PD-L1 inhibitor atezolizumab versus single-agent sorafenib has established a new standard of care for HCC patients with advanced disease. According to IMbrave150, atezolizumab - bevacizumab have reported statistically significant and clinically meaningful benefits in several clinical outcomes, including objective response rate (ORR), progression-free survival (PFS), and overall survival (OS), with these advantages also confirmed by the updated results of this trial, showing a median OS of more than 19 months in HCC patients receiving the immune-based combination. Despite ICI seem to have finally found their role in HCC as part of combinatorial strategies, several questions remain unanswered. Among these, the lack of validated biomarkers of response represents an important issue since only a proportion of HCC patients benefit from immunotherapy. Based on these premises, a greater understanding of the role of potential biomarkers including programmed death ligand 1 (PD-L1) expression, tumor mutational burden (TMB), microsatellite instability (MSI) status, gut microbiota and several others is fundamental. In addition, clinical trials on HCC immunotherapy widely differed in terms of drugs, patients, designs, terms of study phases, and inconsistent clinical outcomes. The background of the changing scenario of medical treatment in HCC should be better discussed, and some recent papers regarding this topic should be included ( PMID: 34976841PMID: 34798793; PMID: 35403533).

Major changes are necessary.

Author Response

Reviewer 2:

The study assesses a current, timely topic in HCC. We recommend some changes:

  1. We believe this article is suitable for publication in the journal although major revisions are needed. The main strengths of this paper are that it addresses an interesting and very timely question and provides a clear answer, with some limitations. The authors should further speculate regarding how this study may impact on clinical practice and its translational implications.

A: Thank you for your positive feedback on our study. We have revised the Introduction section (line 55 – line 69, line 112 – line 115) and Discussion section (line 328 – line 333) to emphasize how this study may impact clinical practice and its translational implications.

  1. A linguistic revision should be performed by a professional service since there are some oversights and grammar mistakes.

A: Thank you for your kind reminder. We have used a paid editing service provided by American Journal Experts (AJE), and the confirmation certificate can be found in the Supplementary Files.

  1. Immune checkpoint inhibitors (ICIs) including pembrolizumab, nivolumab, durvalumab, atezolizumab, etc. have been recently evaluated in HCC patients, and clinical trials assessing single-agent ICI have reported disappointing results. Conversely, immune-based combinations have been more striking. In fact, the phase III IMbrave150 trial assessing the combination of the antiangiogenic agent bevacizumab plus the PD-L1 inhibitor atezolizumab versus single-agent sorafenib has established a new standard of care for HCC patients with advanced disease. According to IMbrave150, atezolizumab - bevacizumab have reported statistically significant and clinically meaningful benefits in several clinical outcomes, including objective response rate (ORR), progression-free survival (PFS), and overall survival (OS), with these advantages also confirmed by the updated results of this trial, showing a median OS of more than 19 months in HCC patients receiving the immune-based combination. Despite ICI seem to have finally found their role in HCC as part of combinatorial strategies, several questions remain unanswered. Among these, the lack of validated biomarkers of response represents an important issue since only a proportion of HCC patients benefit from immunotherapy. Based on these premises, a greater understanding of the role of potential biomarkers including programmed death ligand 1 (PD-L1) expression, tumor mutational burden (TMB), microsatellite instability (MSI) status, gut microbiota and several others is fundamental. In addition, clinical trials on HCC immunotherapy widely differed in terms of drugs, patients, designs, terms of study phases, and inconsistent clinical outcomes. The background of the changing scenario of medical treatment in HCC should be better discussed, and some recent papers regarding this topic should be included (PMID: 34976841; PMID: 34798793; PMID: 35403533).

A: Thank you for your suggestions. Your detailed description of the current situation of immunotherapy for HCC has greatly benefited us. We have revised the Introduction section (line 55 – line 69) to better discuss the background of the changing scenario of medical treatment in HCC and cited relevant papers (ref. 5, 7, 9).

Reviewer 3 Report

In the article entitled “The effect of histone chaperones HSPA8 and DEK on tumor immunity in hepatocellular carcinoma”, the authors aimed to investigate the effect of histone chaperones on tumor immunity in hepatocellular carcinoma (HCC).

They performed bioinformatics analyses to screen immune-related histone chaperones and to divide the HCC samples into two different clusters (based on the expression levels of HSPA8 and DEK). Finally, they analyzed the expressions of immunomodulatory genes, chemokines and chemokine receptorsm in order to evaluate the sensitivity to immunotherapy.

They concluded that, between known histone chaperones and the correlation with the ESTIMATE score, HSPA8 and DEK may be used as biomarkers for precisely predicting the effect of immunotherapy, and that FBLN2 may be used as a therapeutic target of HCC.

Data reported are too preliminary and based on bioinformatics data, only. They may be interesting and taken in consideration for publication only if they are supported by experimental data and analyses. The paper is poorly written and needs to be reorganized in a more simple way, in order to be better understood before it can be considered for publicaton. I reported here my concerns:

The authors should add an ABBREVIATIONS Section to the manuscript in order to clarify the meaning of such acronyms reported within the text.

Many sentences are written in very poor and not understanding way. For example, in the INTRODUCTION Section, the authors reported that “However, due to the complexity of immune system, the molecular mechanisms regulating immune responses and evasion remain unclear which lead to that partial HCC patients present low sensitivity to immunotherapy [6]”. The authors should reorganize these sentences and check for correct english grammar.

In the INTRODUCTION Section, the authors reported that “Then we found that Cluster 2, with high expression of HSPA8 and low expression of DEK, tended to have immune infiltration and better sensitivity to immunotherapy than Cluster 1”. How it is characterized the Cluster 1? Have the authors found samples with high expression of both HSPA8 and DEK or with low expression of HSPA8 and DEK? How did the authors decided the cut-off values for Clusters generation? Please clarify all these aspects of this work.

There are many lacking data, such as:

- “36 known histone chaperones” (lane 93): form where these data came out?

- the four indices of ESTIMATE in TCGA-LIHC samples (Figure 1A) (lane 94): what are these four indices? How many TCGA-LIHC samples were used for the analyses?

- 369 TCGA-LIHC samples (lane 97) are the same of the previous analysis?

In FIGURES 1B-1C-11, they are reported the HSPA8 and DEK transcript expressions for the reported analysis. What about proteins expressions? Please add this data to support the obtained results.

With respect to FIGURES 1D-E, please associate supplementary tables with raw data, in order to organize more clearly the data obtained from these analyses.

Histograms in FIGURES 3-7-8-9-10 are very difficult to read. Please try to reorganize and show them in a more comprehensive way. Also for them, please associate supplementary tables with raw data.

In the RESULTS Section, paragraph named “2.4. Screening of hub genes related to immune infiltration”, the authors reported that “To find out the reason for the difference of immune features between the two clusters, we identified 505 DEGs (197 up-regulated and 308 down-regulated in Cluster 2) (Figure 4A)”. How did they were identified these DEGs genes? Please explain.

From what analysis the “121 genes in the blue module” came out? Please clarify. Also for these, please associate a supplementary table with raw data by listing these genes.

FIGURE 4, in particular the sub-figure 4F is poorly comprehensive, with respect to the text. Please revise it, and reorganize the whole FIGURE 4.

In the RESULTS Section, paragraph named “2.5. Determination of the functions of FBLN2”. From what analysis these data came out?

Furthermore, the data reported by the authors are bioinformatics results regarding possible correlations between the gene but there are any data regarding functional links between them. They seem to be only speculation without strong supporting data.

In the RESULTS Section, paragraph named “2.6. Validation of immune features of the two clusters by GEO and ICGC database”, the authors reported that “As same as previous, we divided 164 HCC samples from GSE19977 [23,24] into two clusters by the expression matrix of HSPA8 and DEK (Figure 7A). Interestingly, Spearman rank correlation analysis showed a weak, negative correlation (R=-0.31, P=6e-05) between the expression of HSPA8 and DEK (Figure 7B). The ESTIMATE, CIBERSORT and ssGSEA analysis showed the similar results with those in TCGA database (Figure 7C-E). Besides, the expression of MHCs, immunomodulatory genes, chemokines and chemokine receptors of the two clusters were evaluated as before (Figure 7F, 8). Meanwhile, we divided 240 HCC samples from ICGC-LIRI-JP into two clusters in the same way, and the remain- ing analyses yielded approximate results (Figure 9, 10). Taken together, Cluster 2 has a feature of stronger tumor immunity than Cluster 1 in TCGA, GEO and ICGC database”. It is not clear how they were analyzed these data. Please verify them and reorganize text and figures.

In the MATERIALS and METHODS Section, sub-paragraph named “4.9. Cell culture and qRT-PCR”, the authors didn’t reported the housekeeping genes(s) used to normalize the results obtained. Please report it and its primers sequences.

In SUPPLEMENTARY FIGURE 1C, it is not reported the mean of y-axis. Please add it.

In conclusion, in this paper, the authors reported some data about the possible effect of histone chaperones on tumor immunity in hepatocellular carcinoma (HCC), and suggest HSPA8 and DEK as biomarkers, and FBLN2 as a therapeutic target of HCC. However data are not always clearly explained both in the text and figures and the conclusions are not supported by them.

I think the bioinformatics data presented need to be supported by experimental functional data by modulating HSPA8, DEK and FBLN2 in HCC cells (in comparison with a normal colon cell line).

The paper needs to be deeply revised, reorganized and improved to meet the journal aims and to be considered for publication.

Author Response

Reviewer 3:

In the article entitled “The effect of histone chaperones HSPA8 and DEK on tumor immunity in hepatocellular carcinoma”, the authors aimed to investigate the effect of histone chaperones on tumor immunity in hepatocellular carcinoma (HCC).

They performed bioinformatics analyses to screen immune-related histone chaperones and to divide the HCC samples into two different clusters (based on the expression levels of HSPA8 and DEK). Finally, they analyzed the expressions of immunomodulatory genes, chemokines and chemokine receptors in order to evaluate the sensitivity to immunotherapy.

They concluded that, between known histone chaperones and the correlation with the ESTIMATE score, HSPA8 and DEK may be used as biomarkers for precisely predicting the effect of immunotherapy, and that FBLN2 may be used as a therapeutic target of HCC.

Data reported are too preliminary and based on bioinformatics data, only. They may be interesting and taken in consideration for publication only if they are supported by experimental data and analyses. The paper is poorly written and needs to be reorganized in a simpler way, in order to be better understood before it can be considered for publication. I reported here my concerns:

  1. The authors should add an ABBREVIATIONS Section to the manuscript in order to clarify the meaning of such acronyms reported within the text.

A: Thank you for your suggestions. We have added an abbreviation list after the Conclusion section (line 573 – line 582).

  1. Many sentences are written in very poor and not understanding way. For example, in the INTRODUCTION Section, the authors reported that “However, due to the complexity of immune system, the molecular mechanisms regulating immune responses and evasion remain unclear which led to that partial HCC patients present low sensitivity to immunotherapy [6]”. The authors should reorganize these sentences and check for correct English grammar.

A: Thank you for your kind reminder. We have used a paid editing service provided by American Journal Experts (AJE) to reorganize these sentences and correct English grammar, and the confirmation certificate can be found in the Supplementary Files.

  1. In the INTRODUCTION Section, the authors reported that “Then we found that Cluster 2, with high expression of HSPA8 and low expression of DEK, tended to have immune infiltration and better sensitivity to immunotherapy than Cluster 1”. How it is characterized the Cluster 1? Have the authors found samples with high expression of both HSPA8 and DEK or with low expression of HSPA8 and DEK? How did the authors decide the cut-off values for Clusters generation? Please clarify all these aspects of this work.

A: Nice questions. Cluster 1 was characterized by low HSPA8 expression and high DEK expression. We have added the characteristics of Cluster 1 to the Introduction section (line 105 – line 106). Consensus clustering is an unsupervised clustering method and a common method for cancer subtype classification. It verifies the rationality of clustering based on the resampling method and automatically selects the optimal number of clusters. Therefore, the authors did not need to set the cutoff values for cluster generation. After the transcriptome profiling data were imported into R software, the gene names (“HSPA8” and “DEK”) were set as clustering criteria. Then, the R package “ConsensusClusterPlus” was used, and consensus clustering was used to divide the HCC samples into different clusters based on the expression matrix of HSPA8 and DEK. The result was a heatmap of the expression of HSPA8 and DEK in each HCC sample, and the samples with the same expression matrix of HSPA8 and DEK were assigned to the same cluster. In our study, 369 HCC samples were automatically classified into two clusters, namely, Cluster 1 with low HSPA8 expression and high DEK expression and Cluster 2 with high HSPA8 expression and low DEK expression. However, no samples were classified as having both high HSPA8 and DEK expression or low HSPA8 and DEK expression. Thank you again for your valuable questions. We have revised the Methods section to clarify this clustering method (line 453 – line 462).

  1. There are many lacking data, such as:
    1. “36 known histone chaperones” (lane 93): form where these data came out?

A: We summarized all the histone chaperones reported in the literature and obtained a list of 36 histone chaperones that have been reported in two papers [Nat Rev Mol Cell Biol, 2017, 18(3):141-158. and Gut, 2020, 69(2):329-342.]. We cited these two papers in our manuscript (line 121, ref. 11 and 13).

  1. the four indices of ESTIMATE in TCGA-LIHC samples (Figure 1A) (lane 94): what are these four indices? How many TCGA-LIHC samples were used for the analyses?

A: Thank you for your questions. ESTIMATE analysis is a method to evaluate the tumor immune microenvironment of each patient with HCC. The four indices include stromal score, immune score, ESTIMATE score and tumor purity. The stromal score and immune score represented the abundance of stromal cells and immune cells, respectively. The ESTIMATE score was calculated based on the stromal score and immune score, representing the extent of immune infiltration. The tumor purity equaled one minus the ESTIMATE score, representing the abundance of tumor cells in tumor tissue. In our study, 369 TCGA-LIHC samples were used for the analysis. We have revised the Methods (line 419 – line 433), Results (line 121) and Figure legends (line 143 – line 144) to clarify the ESTIMATE analysis.

  1. 369 TCGA-LIHC samples (lane 97) are the same of the previous analysis?

A: Yes. We have revised the Methods, Results and Figure legends to clarify the ESTIMATE analysis.

  1. In FIGURES 1B-1C-11, they are reported the HSPA8 and DEK transcript expressions for the reported analysis. What about proteins expressions? Please add this data to support the obtained results.

A: Thank you for your suggestions. We examined the protein levels of HSPA8 and DEK in human HCC cell lines and human normal hepatocytes, and the results showed that HCC cells have higher protein levels of HSPA8 and DEK than normal hepatocytes. We have supplied these data in the new Figure 7 (we have placed the previous Figures 7-10 in the Supplementary Files, so the previous Figure 11 is now the new Figure 7). We also added western blotting-related content to the Results section (line 318 – line 322) and Methods section (line 533 – line 545).

  1. With respect to FIGURES 1D-E, please associate supplementary tables with raw data, in order to organize more clearly the data obtained from these analyses.

A: Thank you for your suggestions. Figure 1D-E was generated by TISIDB. TISIDB (http://cis.hku.hk/TISIDB/index.php) was used to assess the correlation between gene expression and the abundance of tumor-infiltrating lymphocytes (TILs) in HCC samples. TISIDB is a web portal that prestores multiple heterogeneous data types from PubMed, TCGA and other public databases. Once we input a gene name (HSPA8 or DEK), it can automatically generate a heatmap of the abundance of TILs based on the expression of this gene. Due to the confidentiality of its algorithms, we cannot obtain the original data. We have revised the Methods section to make the TISIDB more understandable (line 444 – line 452).

  1. Histograms in FIGURES 3-7-8-9-10 are very difficult to read. Please try to reorganize and show them in a more comprehensive way. Also for them, please associate supplementary tables with raw data.

A: Thank you for your suggestions. According to another reviewer’s comments, we have placed Figures 7-10 in the Supplementary Files so that the number of main figures has been reduced to 7. We have modified the figure legends to make the histograms more understandable (line 186 – line 189). In addition, we have supplied the raw data in the Supplementary Files.

  1. In the RESULTS Section, paragraph named “2.4. Screening of hub genes related to immune infiltration”, the authors reported that “To find out the reason for the difference of immune features between the two clusters, we identified 505 DEGs (197 up-regulated and 308 down-regulated in Cluster 2) (Figure 4A)”. How did they were identified these DEGs genes? Please explain.

A: The differentially expressed genes (DEGs) were defined as genes that had significantly different expression levels between Cluster 1 and Cluster 2, and FoldChange was defined as the ratio of the expression amount of a gene in Cluster 2 to that in Cluster 1. After the transcriptome profiling data were imported into R software, |log2FoldChange| >1 and adjusted P value <0.05 were set as threshold values. Then, the DEGs were identified by the software. We have revised the Methods (line 472 – line 480), Results (line 191 – line 205) and Figure legends (line 230 – line 235) to make this part more understandable.

  1. From what analysis the “121 genes in the blue module” came out? Please clarify. Also for these, please associate a supplementary table with raw data by listing these genes.

A: Nice question. The 121 genes in the blue module were generated from weighted correlation network analysis (WGCNA). After the transcriptome profiling data and the corresponding clinical information were imported into R software, the R package “WGCNA” was used. Then, WGCNA was conducted to divide DEGs into several modules based on their correlations with the immune infiltration indices, including stromal score, immune score, ESTIMATE score and tumor purity. The modules were named with different colors. Then, we found that the blue module, which included 121 genes, had the strongest extent of tumor immunity. We have revised the Results section (line 194 – line 205) and Methods section (line 481 – line 495) to make the WGCNA easier to understand and uploaded the raw data of 121 genes in the blue module in the Supplementary Files.

  1. FIGURE 4, in particular the sub-figure 4F is poorly comprehensive, with respect to the text. Please revise it, and reorganize the whole FIGURE 4.

A: Thank you for your suggestions. We have described this paragraph in more detail (line 190 – line 228) and revised the figure legends of Figure 4 (line 230 – line 235) to make it more understandable. Thank you again for your suggestions.

  1. In the RESULTS Section, paragraph named “2.5. Determination of the functions of FBLN2”. From what analysis these data came out?

A: Thank you for your question. First, we found that among the 12 hub genes, only FBLN2 was associated with prognosis by the Kaplan‒Meier method and log-rank test (Figure 6A). Next, multivariate Cox regression analysis showed that FBLN2 was an independent protective factor in HCC patients (Figure 6B). Then, we found a strong correlation between FBLN2 and the abundance of tumor-infiltrating lymphocytes through TISIDB (Figure 6C). Finally, CIBERSORT analysis and ssGSEA showed that patients with high FBLN2 expression had stronger immune infiltration than those with low FBLN2 expression (Figure 6D, E). We have revised this paragraph (line 236 – line 252) and the Methods section to clarify this point (line 419 – line 510).

  1. Furthermore, the data reported by the authors are bioinformatics results regarding possible correlations between the gene but there are any data regarding functional links between them. They seem to be only speculation without strong supporting data.

A: Nice question. 1. This study established two verification databases in strict accordance with the flow of bioinformatics analysis, and the results generated from verification databases were highly consistent with those from the training database (Part 2.7 of the Results section). 2. We have supplied the results of western blotting and confirmed the conclusions at both the mRNA and protein levels. Of course, we will further explore the immune regulation of HSPA8 and DEK in HCC in future research, and we believe this future study can introduce new breakthroughs to the field of immunotherapy for HCC. Thank you again for your suggestions.

  1. In the RESULTS Section, paragraph named “2.6. Validation of immune features of the two clusters by GEO and ICGC database”, the authors reported that “As same as previous, we divided 164 HCC samples from GSE19977 [23,24] into two clusters by the expression matrix of HSPA8 and DEK (Figure 7A). Interestingly, Spearman rank correlation analysis showed a weak, negative correlation (R=-0.31, P=6e-05) between the expression of HSPA8 and DEK (Figure 7B). The ESTIMATE, CIBERSORT and ssGSEA analysis showed the similar results with those in TCGA database (Figure 7C-E). Besides, the expression of MHCs, immunomodulatory genes, chemokines and chemokine receptors of the two clusters were evaluated as before (Figure 7F, 8). Meanwhile, we divided 240 HCC samples from ICGC-LIRI-JP into two clusters in the same way, and the remaining analyses yielded approximate results (Figure 9, 10). Taken together, Cluster 2 has a feature of stronger tumor immunity than Cluster 1 in TCGA, GEO and ICGC database”. It is not clear how they were analyzed these data. Please verify them and reorganize text and figures.

A: Thank you for your suggestions. The GEO and ICGC databases were used as verification databases to validate previous results from the TCGA database. The analytical methods performed for verification databases were the same as those for the TCGA database. First, HCC samples were divided into two clusters based on the expression matrix of HSPA8 and DEK via consensus clustering. Next, the ESTIMATE, CIBERSORT and ssGSEA analyses showed that Cluster 2, with high HSPA8 expression and low DEK expression, had a higher stromal score, immune score, higher ESTIMATE score and higher expression of immune cell subtypes than Cluster 1, which was characterized by low HSPA8 expression and high DEK expression. In addition, Cluster 2 had higher expression of MHCs, immunomodulatory genes and anti-HCC chemokines and chemokine receptors. Taken together, Cluster 2 is characterized by stronger tumor immunity than Cluster 1 in the TCGA, GEO and ICGC databases. Given space limitations, we did not comprehensively describe Part 2.6 of the Results section in the first submission, but we have clarified this part in the revision (line 265 – line 292).

  1. In the MATERIALS and METHODS Section, sub-paragraph named “4.9. Cell culture and qRT-PCR”, the authors didn’t report the housekeeping genes(s) used to normalize the results obtained. Please report it and its primers sequences.

A: Thank you for the reminder. The housekeeping gene was 18S, and we have added its primer sequences to the Methods section (line 526 – line 528).

  1. In SUPPLEMENTARY FIGURE 1C, it is not reported the mean of y-axis. Please add it.

A: Thank you for your kind reminder. We have added the mean of the y-axis.

In conclusion, in this paper, the authors reported some data about the possible effect of histone chaperones on tumor immunity in hepatocellular carcinoma (HCC), and suggest HSPA8 and DEK as biomarkers, and FBLN2 as a therapeutic target of HCC. However, data are not always clearly explained both in the text and figures and the conclusions are not supported by them. I think the bioinformatics data presented need to be supported by experimental functional data by modulating HSPA8, DEK and FBLN2 in HCC cells (in comparison with a normal colon cell line). The paper needs to be deeply revised, reorganized and improved to meet the journal aims and to be considered for publication.

A: Finally, thank you sincerely for your insightful suggestions, which make our study more convincing. We have thoroughly revised our manuscript based on your comments and hope that you are satisfied with the revision.

Round 2

Reviewer 2 Report

The authors addressed all the queries we raised.

We recommend Acceptance.

Author Response

Reviewer 2:

The authors addressed all the queries we raised. We recommend Acceptance.

A: Thank you very much for your positive feedback of our study and for taking the time to review our manuscript.

Reviewer 3 Report

In the 2nd versione of the article entitled “The effect of THE histone chaperones HSPA8 and DEK on tumor immunity in hepatocellular carcinoma”, the authors responded to my comments.

With respect to my comments, they added interesting and clarifying images to the manuscript, and reorganized many paragraphs and concepts about the showed data. At the end, the paper results improved and clearer than in the first version.

Despite these considerations, I consider that the manuscript needs an experimental validation of such data yet, but if it works well to the Editor I think it may be considered for publication after some little revisions

With regards to such elements to improve the manuscript, I suggest to add some data.

In particular, in the NEW FIGURE 7, please add the molecular weight (kDa) of the revealed proteins to the WB panels and the histograms with the ratio revealed proteins (HSPA8 and DEK)/housekeeping protein (GAPDH).

In FIGURES 1B-1C, the authors didn’t add the HSPA8 and DEK protein expressions in medical samples. Please add this data to the manuscript, in order to support previous analyses.

In the MATERIALS and METHODS Section, sub-paragraph named “Statistical analysis”, the authors didn’t reported the number of experiments the statistical analysis is derived from., with regards to the transcripts/protein analyses. How many replicates did the authors do for each experiment or analysis? The authors reported a statistical analysis and standard deviations in figures, but they didn’t reported how many experiments are conducted in order to confirm the reproducibility of the results obtained. Please insert this data.

I think the paper meets can be considered for publication after these revisions at least

Author Response

Reviewer 3:

In the 2nd version of the article entitled “The effect of THE histone chaperones HSPA8 and DEK on tumor immunity in hepatocellular carcinoma”, the authors responded to my comments.

With respect to my comments, they added interesting and clarifying images to the manuscript, and reorganized many paragraphs and concepts about the showed data. At the end, the paper results improved and clearer than in the first version.

Despite these considerations, I consider that the manuscript needs an experimental validation of such data yet, but if it works well to the Editor, I think it may be considered for publication after some little revisions.

With regards to such elements to improve the manuscript, I suggest to add some data.

  1. In particular, in the NEW FIGURE 7, please add the molecular weight (kDa) of the revealed proteins to the WB panels and the histograms with the ratio revealed proteins (HSPA8 and DEK)/housekeeping protein (GAPDH).

A: Thank you for your suggestions. We have added the molecular weight of the revealed proteins to the WB panels and the histograms with the ratio revealed proteins (HSPA8 and DEK) / housekeeping protein (GAPDH) in Figure 7 (see new Figure 7B).

  1. In FIGURES 1B-1C, the authors didn’t add the HSPA8 and DEK protein expressions in medical samples. Please add this data to the manuscript, in order to support previous analyses.

A: Thank you for your valuable comments. The Human Protein Atlas (HPA) database is a web portal that prestores distribution information of 26000 human proteins in tumor tissue, including HCC tissue. Once we input a protein name (e.g., HSPA8, DEK), we can automatically obtain the immunohistochemical results of this protein in HCC samples and normal liver samples. Then we found that the protein levels of both HSPA8 and DEK were higher in human HCC samples than in normal liver samples (see new Figure 1D and 1E). We have added the relevant content to the Results section (line 126 – line 130), the figure legends of Figure 1 (line 149 – line 150), and the Methods section (line 451 – line 457).

  1. In the MATERIALS and METHODS Section, sub-paragraph named “Statistical analysis”, the authors didn’t report the number of experiments the statistical analysis is derived from., with regards to the transcripts/protein analyses. How many replicates did the authors do for each experiment or analysis? The authors reported a statistical analysis and standard deviations in figures, but they didn’t report how many experiments are conducted in order to confirm the reproducibility of the results obtained. Please insert this data.

A: Thank you for your reminder. The results of PCR and WB experiments are representative of 3 separate experiments. We have declared it in the Figure legend section (line 332).

I think the paper meets can be considered for publication after these revisions at least.

A: Finally, thank you sincerely for your insightful suggestions, which make our study more convincing. And thank your again for taking the time to review our manuscript, hoping that you are satisfied with the revision.
